# Hepatitis C Virus-Lipid Interplay: Pathogenesis and Clinical Impact

**DOI:** 10.3390/biomedicines11020271

**Published:** 2023-01-19

**Authors:** Wesal Elgretli, Tianyan Chen, Nadine Kronfli, Giada Sebastiani

**Affiliations:** 1Division of Experimental Medicine, McGill University, Montreal, QC H4A 3J1, Canada; 2Division of Gastroenterology and Hepatology, Department of Medicine, McGill University Health Centre, Montreal, QC H4A 3J1, Canada; 3Department of Medicine, Division of Infectious Diseases and Chronic Viral Illness Service, McGill University Health Centre, Montreal, QC H4A 3J1, Canada; 4Centre for Outcomes Research and Evaluation, Research Institute of the McGill University Health Centre, Montreal, QC H4A 3J1, Canada

**Keywords:** hepatitis C, lipid metabolism, cholesterol, steatosis, atherosclerosis

## Abstract

Hepatitis C virus (HCV) infection represents the major cause of chronic liver disease, leading to a wide range of hepatic diseases, including cirrhosis and hepatocellular carcinoma. It is the leading indication for liver transplantation worldwide. In addition, there is a growing body of evidence concerning the role of HCV in extrahepatic manifestations, including immune-related disorders and metabolic abnormalities, such as insulin resistance and steatosis. HCV depends on its host cells to propagate successfully, and every aspect of the HCV life cycle is closely related to human lipid metabolism. The virus circulates as a lipid-rich particle, entering the hepatocyte via lipoprotein cell receptors. It has also been shown to upregulate lipid biosynthesis and impair lipid degradation, resulting in significant intracellular lipid accumulation (steatosis) and circulating hypocholesterolemia. Patients with chronic HCV are at increased risk for hepatic steatosis, dyslipidemia, and cardiovascular disease, including accelerated atherosclerosis. This review aims to describe different aspects of the HCV viral life cycle as it impacts host lipoproteins and lipid metabolism. It then discusses the mechanisms of HCV-related hepatic steatosis, hypocholesterolemia, and accelerated atherosclerosis.

## 1. Introduction

Hepatitis C virus (HCV) is a small enveloped virus and has been classified as a *Hepacivirus* genus within the family *Flaviviridae* and separated into several distinct genotypes [1]. The most geographically distributed genotypes are HCV genotypes 1 and 3 [2]. Although all the genotypes can establish chronic infection, genotypes 1 and 3 are associated with specific properties and different clinical presentations. For example, hepatic steatosis (HS) is more prevalent in genotype 3 infections and there is evidence for a direct role for viral factors in the development of this pathology [2,3]. The HCV genome is a positive-sense, single-stranded RNA of ~9600 nucleotides that comprises a 5′-untranslated region (UTR), an open reading frame (ORF), and a 3′UTR (Figure 1). The ORF is translated by the host cell translation machinery into a polyprotein of 3000 amino acids, which is cleaved by viral proteases and cellular peptidase into 10 viral proteins, including three structural proteins and seven nonstructural proteins. The structural proteins, which are located at the N-terminus of the glycoprotein, constitute Core, and two envelop glycoproteins (E1 and E2). The Core forms the capsid shell into which the virus genome is packaged, while the glycoproteins are localized to the lipid envelope surrounding the capsid. The nonstructural proteins (P7, NS2, NS3, NS4A, NS4B, NS5A, and NS5B) lie at the C-terminus of the polyprotein and are critical for viral RNA synthesis, assembly, and other steps of the viral life cycle. P7 is a small protein that resides immediately downstream from E2 and is required for viral assembly and release of viral particles [4,5,6]. It has been demonstrated that, both in vitro and in vivo, the Core protein may have a significant impact on the transcriptional regulation of the lipid regulating factor angiopoietin-like protein-3 (ANGPTL-3). ANGPTL-3 may be considered a persistent HCV molecular fingerprint that may contribute to the continuing dysregulation of lipid metabolism in individual who have been achieved successful eradication of the HCV, as well as, predispose some of them to the development of HCC through core-induced hepatocarcinogenic mechanisms [7,8].

HCV infection is a leading cause of chronic liver disease. Based on the latest estimations of the World Health Organization on HCV in 2019, 58 million individuals are chronically infected globally [9]. The virus is spread through parenteral exposure to an infected person’s blood. The two most common exposures associated with the transmission of HCV are blood transfusion and injection drug use [10]. Acute infection is frequently asymptomatic, with less than 30% of infected people experiencing symptoms such as nausea, weight loss, and, in rare cases, jaundice [11]. HCV stimulates both innate and adaptive immune responses, which are capable of clearing the infection in 15–45% of individuals [12]. Failure to clear the virus results in chronic HCV infection (CHC), which is responsible for HCV liver-related morbidity and mortality including HS, liver cirrhosis, and hepatocellular carcinoma (HCC) [13]. In addition, CHC induces several extrahepatic complications, such as dyslipidemia, cardiovascular disease, type 2 diabetes mellitus, insulin resistance, neurocognitive dysfunction, systemic vasculitis, B cell non-Hodgkin lymphoma, and chronic kidney disease [14,15]. HCV’s tendency to deregulate crucial host functions, primarily innate immunity and lipid metabolism, is what leads to both the persistence and pathogenesis of HCV infection. The relationship between HCV and lipid and lipoprotein metabolism has long been documented clinically. Particularly, chronic HCV is thought to be the cause of dyslipidemia and HS [16]. To date, no effective HCV vaccine is available [4]. Until about ten years ago, combination therapy with pegylated interferon (IFN) and ribavirin was the standard treatment for CHC [17]. The development of direct-acting antivirals (DAA) in the past several years has revolutionized the treatment of CHC due to their superior sustained virological response (SVR) rate, shorter treatment duration, and fewer adverse effects [18]. In this review, we discussed the connection between HCV and host lipid metabolism, as well as the role that lipids play in each stage of the HCV life cycle. Additionally, we will highlight some of the clinical consequences of this interplay in patients with CHC, as demonstrated in several studies (Table 1).

## 2. HCV Life Cycle and Its Link to Lipids

### 2.1. The Unique Composition of HCV Particles

The most remarkable characteristic of infectious HCV particles is their buoyant density, which is both exceptionally low and heterogeneous for an enveloped RNA virus. HCV particles that are separated from the patients’ extracellular compartment are in intimate association with lipoproteins, forming hybrid particles called ‘lipoviroparticles’ (LVP) [19,20]. The LVPs are composed of viral components (a nucleocapsid holding the single-stranded RNA genome that is linked to the viral core protein and envelop membrane comprising surface glycoproteins E1 and E2) as well as different apolipoproteins (e.g., apoE, apoB, apoCI, apoCII and apoCIII) [21] (Figure 2). This results in HCV having low-density characteristics dispersed throughout a broad range of density gradients between 1.03 and 1.20 g/cm^3^ [22]. Furthermore, HCV particles contain lipids and cholesteryl ester contents comparable to those of very-low density lipoproteins (VLDL) and low-density lipoprotein (LDL) [23,24]. The presence of these apolipoproteins influences HCV stability and susceptibility to neutralizing antibodies, enabling the virus to attach to different lipoprotein receptors and enter hepatocytes [19,25].

### 2.2. The Mechanism of HCV Entry into Host Hepatocytes

The life cycle of HCV begins with its entry into cells. It is a highly elaborated multi-step process that requires the interaction of the virus with host cellular molecules [26,27]. The initial step of HCV entry from blood stream to target cell involves the interaction of apoE found on the surface of HCV-LVP with glycosaminoglycans and LDL receptors (LDLr) [28,29]. Under normal physiological conditions, LDLr is responsible for intracellular cholesterol-rich LDL transport through clathrin-mediated endocytosis. Due to the competitive nature between HCV and LDL for LDLr, it has been suggested that a greater level of apoB associated cholesterol such as LDL might predict HCV treatment response [30,31]. Following the completion of the first step, the viral envelop glycoprotein E2 interacts with scavenger receptors class B type I, a high-density lipoprotein (HDL) receptor that binds also VLDL and LDL particles [32,33]. It was shown that this interaction with scavenger receptors class B type I results in E2 glycoprotein conformational changes that enable HCV to interact with tetraspanin CD81 to form HCV-CD81 complex [34]. After the binding to HCV, CD81 moves toward the tight junctions and interacts with claudin-1 and occludin. This interaction leads to cellular internalization of the virus which is mediated through a clathrin-dependent endocytosis process. This movement depends on several regulatory factors including receptor tyrosine kinases, and the Niemann–Pick C1-like protein 1 (NPC1L1) [35,36]. The NPC1L1 is a transmembrane cholesterol uptake receptor that is particularly prevalent in intestinal enterocytes and hepatocytes [37]. The NPC1L1 plays a critical role in viral entry by cholesterol regulation, while receptor tyrosine kinases help HCV entry by regulating CD81-claudin-1 co-receptor associations and membrane fusion. Some studies suggest that receptor tyrosine kinases may be a potential target for prevention and treatment of HCV infection [38]. Another study showed that movement of CD81 toward the tight junction depends on the activation of epidermal growth factor receptor which triggers the actin-mediated lateral membrane diffusion of HCV-CD81 complexes. HCV internalization induces fusion between viral glycoproteins and early endosomes, as well as acidification of the vacuole [39]. Following this pH-dependent process, the HCV capsid is delivered into the cytosol, destroyed and the resultant HCV genomic RNA is ready to be translated to produce viral proteins and initiate viral replication.

### 2.3. HCV Replication, Assembly and Release

Following the release of the HCV genome into the cytosol, viral replication is facilitated by the interaction of HCV with different lipid-related factors. The positive-sense single-stranded RNA is ready to be translated by host ribosomal subunits found on the rough endoplasmic reticulum. Subsequently, the ribosome–RNA complex binds to the endoplasmic reticulum membrane, initiating HCV polyprotein translation. Translation of the HCV genome is largely regulated by a highly conserved structural region called the internal ribosome entry site (IRES) as well as the microRNA-122 binding site, which are located in the 5′ UTR [40,41]. The 5′UTR IRES is responsible for initiation of HCV viral RNA translation by facilitating its binding to the ribosomal subunit. The microRNA-122 is a liver-specific human microRNA which has an important role in HCV viral replication inside the liver cell [42,43]. Interestingly, binding of microRNA-122 to viral RNA results in upregulation of both HCV RNA and genes that are related to plasma cholesterol and hepatic fatty-acid metabolism [44,45]. A single polyprotein of around 3000 amino acids is generated by the translation process which then undergoes proteolytic processing within the rough ER using cellular and viral proteases. The end product is a total of 10 mature HCV proteins, which are comprised of both structural and nonstructural proteins. Following the synthesis of viral proteins, the nonstructural proteins are firmly integrated in or associated with the endoplasmic reticulum membrane through a Geranylgeranyl pyrophosphate-mediated process [45,46,47]. Geranylgeranyl pyrophosphate is the product of cholesterol biosynthetic pathway and its role in viral protein-membrane association is largely dependent on the fatty acid contents of the cell. Inhibition of fatty acid synthesis leads to the inhibition of HCV viral replication [45,48]. The viral NS4B and NS5A stimulate cellular lipid lipase to induce alterations in endoplasmic reticulum membrane architecture to form a cluster of cholesterol-rich double-membrane vesicles which are associated with intracellular lipid droplets (LD) [49,50,51]. This forms the so-called membranous web which is used as a place for viral replication. Several lipid transfer proteins such as Niemann–Pick C1protein, which is responsible for transporting LDL-derived cholesterol, are suggested to play a crucial role in recruitment of cholesterol to the membranous web. Pharmacological inhibition of Niemann–Pick C1protein was associated with a decrease in cholesterol at replication sites leading to reduced HCV viral replication [52]. Within the membranous web, nonstructural proteins act as a replication complex which is responsible for replication of the newly synthesized viral RNA. In the procedure catalyzed by NS5B, the RNA-dependent RNA polymerase, the positive RNA genome serves as the template for the negative HCV RNA strands. Consequently, the newly generated strands serve as templates for the synthesis of positive HCV RNA strands. Parallel to the creation of new RNA strands, new viral proteins are created during the translation process. After the formation of positive HCV RNA strands and viral structural proteins, the assembly of new HCV particles can begin.

To initiate virion assembly, it is believed that replicated genomes must be released from the membranous web to make contact with the core protein that forms the virion capsid. Several studies have identified the LD and VLDL biogenesis pathways as major contributors from the host cell to HCV assembly [53]. Initial stages of HCV assembly are assumed to require the association between the core protein and LD. LD are cytosolic storage organelles that are produced in endoplasmic reticulum and are composed of triglycerides and cholesterol esters surrounded by phospholipid monolayers with a variety of proteins on its surface [54]. The core protein is composed of two domains (D1 and D2), and it is the D2 domain that mediates the core–LD interaction. Mutation in D2 leads to disruption in the core–LD association which results in deceased infectious HCV production [55,56]. The ability of the core to localize to LD is believed to be largely dependent on host diacylglycerol acyltransferase 1, an enzyme involved in synthesis of triglycerides in the endoplasmic reticulum as well as LD and VLDL morphogenesis [57]. As it is localized to the LD, this allows the core protein to recruit newly synthetized HCV RNA from the membranous web and envelope E1 and E2 proteins from ER [58]. The NS5A protein is believed to facilitate the transport of viral RNA to the LD to be encapsulated by the core protein. This process require the interaction of NS5A with the D1 domain of the core protein [59]. Next, the core–capsid complex migrates to the endoplasmic reticulum membrane where it interacts with viral E1/E2 proteins to form the envelop which is acquired by budding into the endoplasmic reticulum at lipoprotein locations where lipidation may take place via interaction between the virion and lipoproteins. HCV assembly has been linked to components of the VLDL synthesis and secretion pathway such as microsomal triglyceride transfer protein (MTP) [60], apoB [61], and apoE [62,63,64]. The first step in the formation of VLDL requires the lipidation of apoB-100 that is mediated by MTP to generate a pre-VLDL particle [65]. Pre-VLDL is subsequently fused with triglyceride-rich droplets to form VLDL [66]. The incorporation of apo-E and apo-CIII on the surface of LD seems to be mediated by MTP inhibitors [67]. It has been shown that the MTP inhibitors have a greater impact on HCV secretion than that of VLDL [64]. It is important to note that HCV replication complexes isolated from human hepatoma cells include all the proteins necessary for VLDL assembly [60]. The mature HCV is linked with VLDL and released through the VLDL secretion pathway as LVP.

## 3. HCV–Lipid Interaction: Clinical Phenotypes

### 3.1. Lipid Profile/Circulating Hypocholesterolemia

Plasma lipid levels are changed in patients with HCV, regardless of the duration of infection. Both acute [68] and chronic HCV infection are associated with reduced levels of circulating LDL, apoB100 and total cholesterol, compared to healthy controls [69,70]. Reduced apoB100 levels and HCV viral loads have been shown to be inversely related in people with non-genotype 1 infection [70]. It is well established that HCV possesses a mutual relationship with host lipid and lipoprotein metabolism [71]. HCV affects multiple mechanisms of lipid metabolism within hepatocytes: it increases lipid biosynthesis, hinders mitochondrial oxidation and consequently lipid degradation, and lowers the export of apolipoproteins—particularly VLDL, leading to significant intracellular lipid accumulation and circulatory hypocholesterolemia and hypolipoproteinemia [72]. The presence and degree of hypocholesterolemia have significant implications on the prognosis of CHC patients. Higher LDL and HDL levels are related with higher SVR rates [31,73]. This may be attributed to the reliance of HCV on LDL cholesterol levels and the LDLr for both cellular entry as well as viral replication.

Following viral eradication through spontaneous clearance of acute infection or successful antiviral treatment, cholesterol and β-lipoprotein levels return to pre-HCV infection levels, implying that HCV gene expression may be responsible for their altered levels [74]. This increase of serum lipids has been described at different magnitudes in the literature [75]. A retrospective study involving HCV-infected patients who were treated and cured with sofosbuvir-based regimens resulted in a significant increase in LDL and total cholesterol up to 6 months post-HCV eradication (LDL: 99.5 ± 28.9 mg/dL vs. 128.3 ± 34.9 mg/dL, *p* < 0.001; total cholesterol: 171.6 ± 32.5 mg/dL vs. 199.7 ± 40.0 mg/dL, *p* < 0.001) [76]. Lacerda et al. found that 1-year post-treatment levels of total cholesterol, LDL, VLDL and triglycerides were significantly elevated in patients who had achieved SVR [77]. Similarly, a study by Driedger et al., carried out on 442 HCV patients who were treated with DAA, found that the level of serum cholesterol and triglycerides were also elevated after achieving SVR [78]. The role of HCV genotype on post-SVR hypercholesterolemia has been also investigated. It has been reported that increased levels of serum cholesterol were associated with genotype 3 in patients who achieved SVR [79]. However, in both genotype 3 non-responders and genotype 1, unchanged serum cholesterol values were observed regardless of the response [79]. Despite the fact that the correction of hypolipidemia has been reported only in genotype 3 in this study, there is accumulating evidence demonstrating that the reversal of hypolipidemia is not HCV-genotype specific [80,81].

Numerous studies suggest that effective HCV eradication may result in increased atherosclerotic cardiovascular disease due to the unfavorable lipid profile that results from reversed hypolipidemia, as evidenced by high serum LDL and small dense LDL, and the latter is a better cardiovascular disease marker than LDL with a higher atherogenic potential. In a case–control study that included 179 patients, Corey et al., found that 13% of the population they evaluated had post-SVR LDL levels that required lipid lowering treatment. These patients had values >130 mg/dL and two or more significant risk factors for coronary artery disease. However, none of these patients had LDL values requiring treatment prior to antiviral therapy [82]. Therefore, deterioration of the lipid profile following treatment may reach a clinically significant level necessitating consideration of cholesterol-lowering medication.

### 3.2. Hepatic Steatosis

HS refers to the excessive accumulation of intrahepatic fat mainly in the form of triglycerides that represents at least 5% of liver weight [83]. It is a hallmark of HCV infection, with its specific clinical and prognostic implications, which can be detected in up to 70% of infected individuals [84]. Prior to the discovery of HCV, the presence of fatty changes in liver biopsies of non-A, non-B hepatitis patients was recognized as a significant characteristic [85]. Once HCV diagnostic tools were available, it was shown that HS is a common histological finding of CHC [86]. Genotype-3 HCV infected patients have the highest prevalence of HS, which can improve and even disappear following successful antiviral therapy [87]. While some lipid accumulation may actually be hepatoprotective, prolonged lipid accumulation can promote the activation of inflammatory responses and the reduction of metabolic competence [88]. HS has become a significant issue in HCV due to its damage on a chronically diseased liver, and is associated with more severe histological damage and higher fibrosis scores in HCV patients, which may indicate that liver fat is a biologically active tissue. HS can be strongly and independently correlated with the degree of fibrosis in CHC [89]. Furthermore, it has been demonstrated that the frequency, severity of HS, and response to antiviral treatment vary according to HCV genotype, suggesting the substantial role of HCV proteins in the accumulation of triglycerides within hepatocytes [87]. Although there have been several publications on the processes that may be involved, the mechanisms underlying the development of HS in the setting of HCV infection remain poorly understood. Both viral and host factors are involved in the development of HS; HCV non-3 genotype-associated steatosis, also called “metabolic steatosis”, is frequently associated with insulin resistance, increased body mass index and visceral obesity. On the other hand, HCV genotype 3-associated steatosis or “viral steatosis” is induced by viral proteins and directly related to viral load [83,89,90,91]. The direct steatogenic role of HCV genotype-3 has been demonstrated in several studies. Castéra et al. have observed a remarkable improvement in HS in patients infected with HCV genotype 3, who achieved SVR using IFN-based regimen [92]. Similar observations were documented in another longitudinal study in which HS was significantly reduced after achieving SVR in patients with HCV genotype 3, but not 1 [93].

HCV utilizes lipid pathways to its own advantage. HCV might have a direct role in inducing lipid accumulation within hepatocytes. HCV depends on intracellular LDs for the accumulation of viral proteins and packaging of viral genomes. Viral proteins, such as Core and NS5A, have been shown to induce the production and accumulation of LDs. Furthermore, a potential effect of viral proteins on the VLDL secretory pathway may be brought about by their inhibitory effect on the MTP, an enzyme that is required for the assembly of VLDL, resulting in the accumulation of triglycerides inside the liver cells, and subsequently, the development of HS [94]. In addition, the HCV core protein may cause mitochondrial dysfunction that can lead to an accumulation of reactive oxygen species and the inhibition of specific antioxidant mechanisms, which may be the mechanisms driving the development of severe oxidative stress in HCV infection. It has been shown that the HCV core protein may lead to decreased expression of peroxisome proliferator-activated receptor alpha, an important regulator of fatty acids degradation in the liver, which can lead to the development of HS [95].

Insulin resistance (IR) may also play an important role in the development of HS in patients infected with HCV. Several studies have been done on the pathophysiology of the HCV-mediated mechanisms that cause IR. In HCV infection, IR may result from increased free fatty acids, increased levels of both suppressor of cytokine signaling 3 and tumor necrosis factor alpha (TNF-α) [96]. These in turn may result in downregulation of insulin receptor substrate signaling1 and hence IR. As a result of IR, glucose accumulation can lead to increased insulin production. Both the abundance of lipogenic substrates (glucose and free fatty acids) and high lipogenic hormone levels (hyperinsulinemia) lead to overstimulation of lipogenesis, eventually resulting in HS (Figure 3) [97].

The relationship between the presence of HS and progression of fibrosis has been well-documented. Afsari et al. examined the risk factors contributing to progression of CHC infection to fibrosis in an African American population. The study showed that HS was independently associated with fibrosis progression [98]. Similarly, an Italian cohort study of 180 patients demonstrated that HS is a significant cofactor in accelerating the process of hepatic fibrosis in HCV patients [99]. Although the association between HS and fibrosis progression in HCV infections has been more thoroughly researched, several studies have shown contradictory findings. In other studies, liver fibrosis progression was linked to age, obesity, elevated serum alanine aminotransferases, and periportal necroinflammation but to a lesser extent with HS [100,101]. Moreover, patients with CHC and viral-induced HS may have worse hepatic outcomes prior to treatment [84]. A recent study indicates that the occurrence of post-SVR HS may not be associated with a lower risk profile [102]. In this study, which looked at the influence of HS on HCC and all-cause mortality in CHC patients following SVR, the presence of fatty liver was linked with a significant 7.5-fold increase in both outcomes (hazard ratio 7.51, 95% confidence interval 3.61–13.36, *p* < 0.001). In addition, there is a substantially increased risk of extrahepatic manifestations, notably cardiovascular disease, which is attributed to both HCV itself and HCV-related HS [103]. These findings underscore the significance of HS as a potential substantial risk factor for poor outcomes and suggest that screening and follow-up in this population require special consideration.

### 3.3. Accelerated Atherosclerosis

Over the two last decades, a strong association between HCV and atherosclerosis has been reported. In 2002, Ishizaka and colleagues first showed a correlation between HCV and the development of carotid artery plaques in a cohort of 4784 Japanese individuals who had undergone general health-screening tests [104]. Since then, a considerable number of cohort studies, systemic reviews, and meta-analyses have attempted to clarify the relationship between HCV infection and cardiovascular disease. This hypothesis was supported by an Italian study comparing patients with chronic liver diseases of viral and nonviral etiologies to healthy controls. This study revealed that intima–media thickness (IMT) of CHC patients was smaller compared to IMT of nonalcoholic fatty liver disease, but larger in comparison to the IMT of controls [105]. Other studies, from both Western and Eastern populations, have further supported this notion. This association was maintained even after adjusting for well-known cardiometabolic covariates such as age, sex, BMI, visceral obesity, hypertension, smoking, IR, diabetes, and metabolic syndrome [103,104,106]. Petta et al., conducted a meta-analysis of nine case–control studies to investigate the impact of HCV infection on carotid plaque. They found a twofold higher risk of carotid plaques in patients infected with HCV compared to controls, without significant heterogenicity within studies [107].

Apart from carotid atherosclerosis, accumulating data suggest that HCV infection can also increase coronary artery disease as well as cerebrovascular disease and stroke. It has been shown that HCV infection enhances the risk of coronary artery disease. Lin et al. found that HCV patients had 1.76-fold increased chance of having ischemic changes on electrocardiogram compared to noninfected individuals [108]. In another study, Butt et al. reported that despite the presence of favorable lipid profile in patients infected with HCV, HCV infection was associated with greater risk of coronary artery disease with a hazard ratio of 1.27 (95% CI, 1.22–1.31) after adjustment of conventional risk factors [109]. Interestingly, a recent case–control study found contradicting results, in which there was no statistically significant association between HCV infection and acute coronary syndrome in the study population [110]. Several studies have examined the potential association between HCV infection and cerebrovascular events. These studies found contradictory findings as well, ranging from HCV infection as a risk factor to HCV infection conferring a protective role [111,112,113,114]. In a population-based cohort study, Liao and colleagues compared the incidence of stroke in newly diagnosed HCV patients and randomly selected controls, age- and sex-matched, from the same database. HCV infection was independently associated with stroke occurrence after adjusting for traditional cardiometabolic risk factors [111]. On the other hand, a case–control study of 126,926 HCV patients and 126,926 controls, showed a significantly lower prevalence and odds of cerebrovascular events in individuals infected with HCV [114]. Due to the presence of conflicting and inconclusive literature, this association warrants further investigation.

Various direct and indirect HCV pro-atherogenic biological processes have been postulated (Figure 4). HCV infection induces hepatic and systemic inflammation by increasing the levels of pro-atherogenic chemokines and cytokines [115] and HS [103], a distinguishing feature of this infection. Boddi et al. showed that HCV colonizes and replicates within carotid plaques, through the isolation of HCV RNA sequences, which in turn may allow the virus to play a direct proatherogenic role by inducing arterial inflammation, likely through the pro-inflammatory cytokine interleukin1β [116]. In addition, HCV structural and non-structural proteins play a significant role in initiating and maintaining chronic inflammation. HCV produces an unbalanced ratio of T helper (Th)1/Th2 cytokines, which disrupts the equilibrium between cellular immunity, promoted and maintained by interleukin (IL)-2, TNF-α and interferon-γ, and humoral immunity, sustained by IL-4, IL-5, IL-6 and IL-10 [117]. Furthermore, HCV infection is associated with elevated levels of endotoxemia and an increased oxidative stress level, which stimulates a robust inflammatory response via TNF- and toll-like receptors, both of which can induce a pronounced inflammatory response. HCV is also responsible for mixed cryoglobulinaemia, a classic vasculitic disease [115].

HCV is considered a metabolic virus and is linked to metabolic illnesses, particularly IR and type 2 diabetes, both of which strongly promote atherogenesis. The higher prevalence of atherosclerosis reported in HCV patients is attributed to HS, which is associated with various pro-atherogenic factors such as inflammatory cytokines, hyper-homocysteinaemia, hypo-adiponectinaemia, IR, and components of the metabolic syndrome [103].

Several studies have investigated whether the use of DAA and HCV eradication improve atherosclerosis. In a large multi-center prospective study of patients with HCV infection with compensated cirrhosis treated with IFN or DAA, the risk of subclinical atherosclerosis and cardiovascular disease was lower in patients who achieved SVR compared with nonresponders [118]. In a large cohort of HCV-infected patients, Butt et al. examined the incidence of cardiac events in treated patients (pegIFN/ribavirin or DAA) matched for age, race, sex, and baseline values with patients who had not previously received treatment. Lower rate of incident cardiac events was reported in the treated group compared to the control group (7.2% vs. 13.8%), and lower incidence of cardiac events (13%) was associated with SVR [119]. In another study, the risk of carotid atherosclerosis was significantly elevated in HCV-infected individuals, but not in HCV-cleared patients, compared to uninfected controls [120]. Similarly, a Taiwanese study of 23,665 individuals reported that the risk of deadly cerebrovascular events progressively increased from patients who were anti-HCV-positive with undetectable HCV RNA, to patients with a low viral load, and further to those who were highly viremic, compared to anti-HCV-negative patients [121]. Of note, other than the direct treatment effect on atherosclerotic risk, therapeutic changes on other cardiometabolic risk factors may also play role in the development of atherosclerosis. Reversal of the lipid profile following HCV eradication has been found to be associated with an increased risk of cardiovascular events. Huang et al. has found that dyslipidemia occurs after HCV eradication, particularly post-SVR LDL-Cholesterol surge of > 40%, was associated with the risk of the vascular events with a hazard ratio of 15.44 (95% CI, 1.73–138.20) [122]. Despite what has been published in the literature on pre- and post-treatment effects on atherosclerosis, all findings must be interpreted with caution. The data also highlight the significance of early treatment in ameliorating proatherogenic metabolic, inflammatory, and immunological factors, thereby improving atherosclerosis.

## 4. Conclusions

Our knowledge of the HCV life cycle has evolved over the last 30 years. In this review, we highlighted the intimate link between each stage of the HCV life cycle and host lipid metabolism. Its interaction with lipoproteins and ability to upregulate hepatic lipid synthesis, impair lipid degradation, and decrease lipoprotein exportation result in significant intracellular lipid accumulation, HS, as well as a relative circulating hypocholesterolemia. Furthermore, patients with CHC infection are at increased risk of accelerated atherosclerosis. The effect of the successful eradication of HCV infection using DAA on the reversal of HCV metabolic manifestations and the development of atherosclerosis is still controversial; more studies are required to better understand the impact.

**Table 1 biomedicines-11-00271-t001:** Summary of studies investigating metabolic manifestations of chronic hepatitis C.

Ref.	The Metabolic Outcome of Interest	Findings
Adinolfi et al. [99], 2001	HS and hepatic fibrosis score	HS, and especially the higher grades, is of clinical importance because it showed a higher hepatic fibrosis score than those with a lower grade or without HS.
Kumar et al. [93], 2002	Post-SVR effect on HS	In individuals with HCV genotype 1, regardless of treatment response, there was no change in HS after treatment. SVR significantly reduced HS among individuals infected with genotype 3, whereas there was no improvement in HS among those without an SVR.
Perlemuter et al. [94], 2002	HS mechanism	Hepatitis C virus core protein inhibits MTP activity and VLDL secretion
Ishizaka et al. [104], 2002	Carotid plaque and carotid IMT	There is an association between HCV seropositivity and carotid-artery plaque and carotid IMT was independent of other atherosclerosis risk factors.
Asselah et al. [101], 2003	HS, necroinflammation, and fibrosis	HS does not appear to be a significant predictor of liver fibrosis in people with CHC. A high stage of fibrosis is related with a high grade of necroinflammation.
Hézode et al. [91], 2004	HS and HCV genotype	In patients infected with HCV genotype 3, the severity of HS was independently related to HCV RNA load alone, whereas in individuals infected with HCV genotype 1, it was independently associated to body mass index, total alcohol intake, and histopathologic activity grade (but not viral load).
Castéra et al. [92], 2004	HS and effect of SVR	There is a remarkable improvement in HS in subjects infected with HCV genotype 3, who acquired SVR.
Perumalswami et al. [100], 2006	HS and fibrosis	In patients with CHC, HS was not associated with the presence of or subsequent progression of fibrosis.
Fernández-Rodríguez et al. [79], 2006	HS and cholesterol level in HCV genotype 3	Besides producing HS, HCV genotype 3 specifically reduces serum cholesterol which then reversed with SVR.
Targher et al. [105], 2007	Carotid IMT	IMT of CHC patients was smaller compared to IMT of NAFLD, but larger in comparison to the IMT of controls
Butt et al. [114], 2007	Coronary artery disease and stroke	The prevalence and likelihood of coronary artery disease and stroke are reduced in HCV-infected individuals.
Reddy et al. [90], 2008	HS and HCV genotype	HS is substantially more prevalent in patients infected with HCV genotype 3 than other genotypes and that successful treatment of HCV genotype 3 infection with interferon plus ribavirin is associated with clearance of HS.
Sheridan et al. [31], 2009	Serum lipid and treatment outcome	Higher apoB-associated cholesterol is associated with improved treatment outcomes in CHC patients receiving antiviral medication.
Corey et al. [82], 2009	post- treatment hyperlipidemia and coronary artery disease	HCV is linked to lower levels of cholesterol and LDL. This hypolipidemia reverse with successful HCV treatment but persists in non-responders. A considerable proportion of successfully treated patients experience LDL and cholesterol rise to levels associated with increased coronary disease risk
Butt et al. [109], 2009	Coronary artery disease	HCV infection is associated with a greater risk of coronary artery disease, even after adjustment for traditional risk factors.
Harrison et al. [73], 2010	Serum lipid, statin use and treatment outcome	Elevated LDL or low HDL levels at baseline, as well as anticipatory statin use, were associated with greater SVR rates.
Mostafa et al. [120], 2010	Lipoprotein profile, insulin resistance, Carotid IMT	The risk of carotid atherosclerosis was considerably elevated in HCV-infected persons, but not in HCV-cleared individuals, compared to uninfected controls.
Lee et al. [121], 2010	Cerebrovascular diseases	HCV infection is linked with a greater risk of cerebrovascular mortality, especially for those with higher serum HCV RNA levels.
Corey et al. [68], 2011	Hypolipidemia	Acute HCV infection results in hypolipidemia characterized by lower LDL, cholesterol and non-HDL cholesterol levels that reverse following infection resolution.
Adinolfi et al. [103], 2012	Carotid atherosclerosis, HS	CHC patients had a greater prevalence of carotid atherosclerosis than controls.CHC with HS had a higher prevalence of carotid atherosclerosis than NAFLD patients.
Petta et al. [106], 2012	Carotid atherosclerosis	Severe hepatic fibrosis is associated with an increased risk of early carotid atherosclerosis in HCV genotype 1 patients.
Liao et al. [111], 2012	Stroke	CHC infection was independently associated with increased risk of stroke after controlling for conventional stroke risk factors.
Lambert et al. [70], 2013	Serum lipids abnormalities	Patients with HCV have increased lipogenesis but decreased cholesterol production compared to healthy people.
Hsu et al. [112], 2013	Stroke and Post-SVR	Interferon-based therapy may decrease stroke risk in CHC patients, independent of other confounders.
Adinolfi et al. [113], 2013	Ischemic stroke.	HCV infection is associated with an increased risk of ischemic stroke.
Chang et al. [80], 2014	Post-SVR changes in cholesterol, triglycerides, HDL, LDL, ApoB.	Significant post-therapeutic elevation in cholesterol, triglyceride, HDL, LDL, ApoA1 and ApoB were reported in patients with SVR but not in those without. Although the pre-treatment lipid profiles of genotype1 and genotype 2 patients were indifferent, genotype 1 had greater post-treatment triglyceride/HDL ratios and triglyceride levels than genotype2 individuals.
Lin et al. [108], 2014	Ischemic heart disease	HCV seropositivity was an independent risk apart from conventional factors for ischemic heart disease.
Foka et al. [7], 2014	SVR, Persistent lipid abnormalities	It has been shown that ANGPTL-3 modulation by the viral core protein through a process that may be linked to deregulated glycolysis, enhanced hepatic lipogenesis, and a steatotic phenotype in hepatocellular adenomas may contribute to persistent HCV manifestations and the development of HCC.
Morales et al. [76], 2016	Sofosbuvir and lipid profile alterations	Post-therapy, eradication of HCV with a Sofosbuvir regimen led to a significant increase in LDL and total cholesterol.
Afsari et al. [98], 2017	HS and fibrosis	HS is independently related to fibrosis in patients infected with HCV.
Lacerda et al. [77], 2018	Lipid changes post-SVR	After an effective treatment, CHC patients experienced a restoration of lipid metabolism.
Cacoub et al. [118], 2018	Subclinical atherosclerosis and cardiovascular disease	SVR was associated with a decreased rate of major cardiovascular events.
Driedger et al. [78], 2019	Serum cholesterol and triglycerides changes after SVR	Serum cholesterol and triglycerides levels were elevated after achieving SVR
Peleg et al. [102], 2019	HS and advanced fibrosis	HCV patients with HS did not have increased incidence of advanced fibrosis or cirrhosis.
Butt et al. [119], 2019	Cardiovascular events and DAAs	There is a substantial benefit of HCV treatment on the incidence and risk of future cardiovascular events.
Huang et al. [122], 2020	Lipid profile, cardio-cerebrovascular events	Serum lipid levels were augmented after HCV eradication, and it was associated with increased risk of cardio-cerebrovascular diseases.
Wu et al. [110], 2021	Acute coronary syndrome	There was no statistically significant correlation between HCV infection and hospitalization for acute coronary syndrome.
Valiakou et al. [8], 2021	SVR and persistent lipid abnormalities	In individuals with advanced liver fibrosis who attained viral clearance in vitro, the levels of ANGPTL-3 remained unaltered. As a result, ANGPTL-3 may be a component of the residual HCV molecular fingerprint, which may contribute to the ongoing dysregulation of lipid metabolism in those who have been cured and may predispose some of them to development of HCC.

ANGPTL-3: Angiopoietin-like protein-3; CHC: Chronic hepatitis C; HCV: Hepatitis C virus; HS: Hepatic steatosis; IMT: Intima-media thickness; HDL: High density lipoprotein; LDL: Low density lipoprotein, VLDL: Very low density lipoprotein; SVR: Sustained virological response.

## Figures and Tables

**Figure 1 biomedicines-11-00271-f001:**
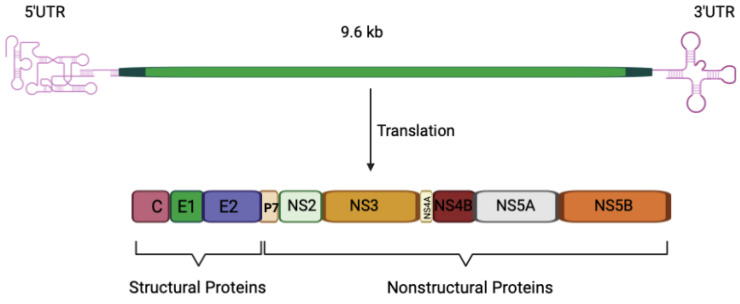
Schematic presentation of HCV genome and polyprotein precursor. The HCV genome encodes a polyprotein of about 3000 amino acids and comprises an open reading frame flanked by 5′ and 3′ untranslated regions (UTRs). Following translation, the core, E1, and E2 proteins, as well as p7, are separated from the polyprotein by cellular peptidase. NS2’s protease activity separates NS2 from NS3, whereas the viral replication components (NS3-NS5B) are separated by the NS3-4A protease. C: core protein; E1 and E2: envelope glycoprotein E1 and E2; NS: nonstructural.

**Figure 2 biomedicines-11-00271-f002:**
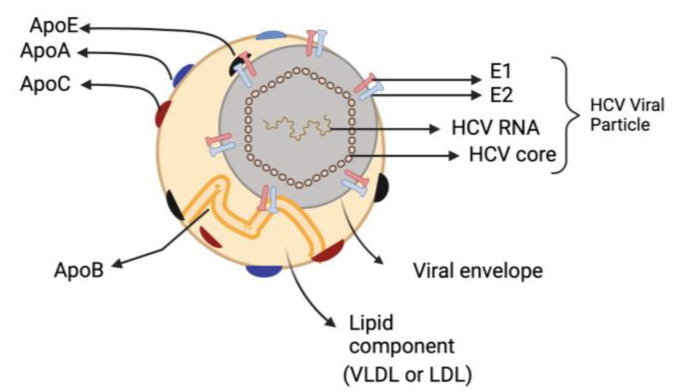
Hepatitis C virus Lipoviroparticles (LVP). The highly infectious HCV particle corresponds to a hybrid particle composed of VLDL or LDL components and viral components named LVP. VLDL: very-low density lipoproteins; LDL: low-density lipoprotein.

**Figure 3 biomedicines-11-00271-f003:**
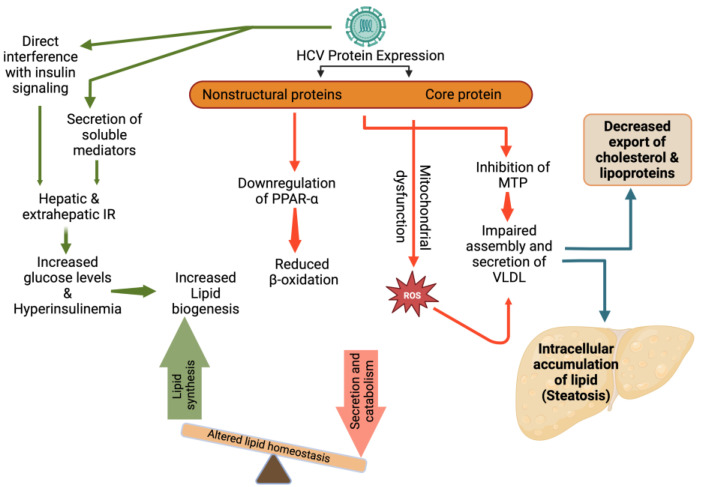
HCV induced alterations in lipid metabolism and steatosis. IR: Insulin resistance; PPAR-α: Peroxisome proliferator-activated receptor-α; ROS: Reactive oxygen species; MTP: Microsomal triglyceride transfer protein; VLDL: very-low density lipoproteins.

**Figure 4 biomedicines-11-00271-f004:**
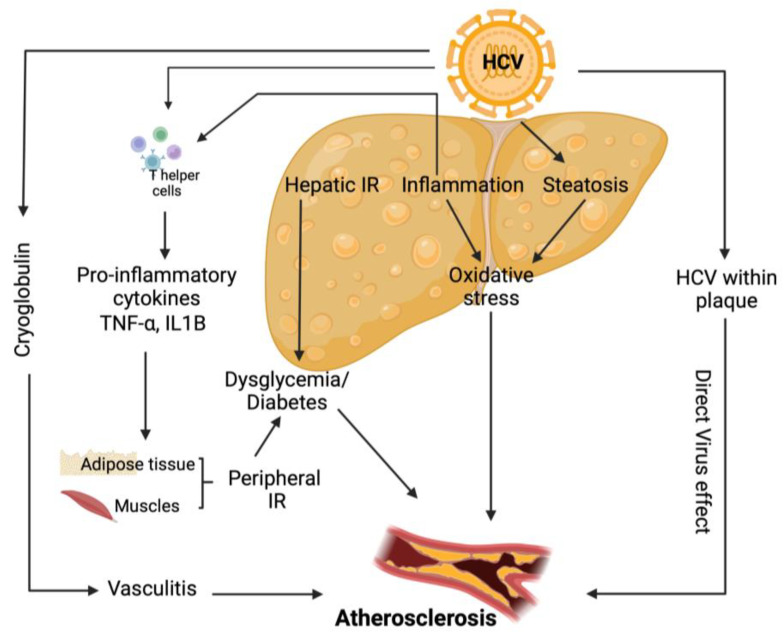
Direct and indirect pathogenic mechanisms responsible for the development of atherosclerosis in chronic hepatitis C infection. HCV: Hepatitis C virus; IR: Insulin resistance; TNF-α: Tumor necrosis factor-α; IL1B: interleukin-B1.

## Data Availability

This study was a review article, so no patient data was collected.

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
