# Peer review of "Hepatitis C Virus-Lipid Interplay: Pathogenesis and Clinical Impact"

_biomedicines, 2023, doi:10.3390/biomedicines11020271_

Round 1
Reviewer 1 Report
The present review by Elgretli and Sebastiani deals comprehensively with the relationship between chronic HCV infection and the alterations in lipid metabolism, including biological mechanisms and clinical implications. In my opinion, the paper is wenn designed and superbly written, so that even a not dedicated professional can follow easily. References are adequate and updated.
I have just some minor suggestions:
- I would suggest and additional figure in the chapter 3.2 ("Hepatic Steatosis"), which might include all the reported mechanisms between HCV and steatosis. I would suggest something like Figure 3 for atherosclerosis, in order to make the contents of the chapter more accessible.
- "Niemann-Pick C1-like protein 1" should be reported the first time and then abbreviated in NPC1L1 in the rest of the manuscript.
- Page 2; raws 57-58: why is the text in Italic? If there is not a specific reason, please correct.
Author Response
Thank you for the insightful comments and feedback that helped us improve this manuscript!
We went through your suggestions and comments and responded accordingly.
Point 1: We incorporated a figure describing the pathogenesis of HCV-induced steatosis (see Figure 3).
Point 2: We abbreviated each consecutive Niemann-Pick C1-like protein 1 as NPC1L1.
Point 3: We corrected this typo that was made while updating the version of the previous Word document.
Reviewer 2 Report
The manuscript of Wesal Elgretli and Giada Sebastiani deals with a very interesting subject concerning HCV pathogenesis. It describes different aspects of the hepatitis C viral life cycle as it impacts host lipoproteins and lipid metabolism. In addition, it discusses the mechanisms of HCV-related hepatic steatosis, hypocholesterolemia, and accelerated atherosclerosis.
Minor comments:
Figure 2: viral envelop should be viral envelope
Figure 3: It will be more representative if authors could enrich the part of the figure dealing with chronic inflammation by introducing liver or other tissues that are affected
It will be of added value to add some references in the table concerning the role of ANGPTLs and in particular ANGPTL-3 that may be part of the residual HCV molecular fingerprint that could contribute to persistent deregulation of lipid metabolism in cured individuals and predispose some of them to HCC development (Foka P. et al., 2014 and Valiakou V. et al., 2021)
Author Response
Thank you for your insightful comments and feedback that helped us improve this manuscript!
We went through your suggestions and comments and responded accordingly.
Point 1: We corrected the typo made during the preparation of the figure. (Please see updated Figure 2.)
Point 2: We updated the figure related to chronic inflammation. (Please see updated Figure 4.)
Point 3: We found these findings very interesting, and we incorporated them both in the introduction section and the table.